# STEM-POM: Evaluating Language Models Math-Symbol Reasoning in Document Parsing

**Jiaru Zou**
University of Illinois at Urbana-Champaign
Champaign, IL
jiaruz2@illinois.edu

**Qing Wang**
University of Illinois at Urbana-Champaign
Champaign, IL
qingw3@illinois.edu

**Pratyush Thakur**
University of Illinois at Urbana-Champaign
Champaign, IL
pthakur3@illinois.edu

**Nickvash Kani**
University of Illinois at Urbana-Champaign
Champaign, IL
kani@illinois.edu

## Abstract

Advances in large language models (LLMs) have spurred research into enhancing their reasoning capabilities, particularly in math-rich STEM documents. While LLMs can generate equations or solve math-related queries, their ability to fully understand and interpret abstract mathematical symbols in long, math-rich documents remains limited. In this paper, we introduce STEM-POM, a comprehensive benchmark dataset designed to evaluate LLMs' reasoning abilities on math symbols within contextual scientific text. The dataset, sourced from real-world ArXiv documents, contains over 2K math symbols classified as main attributes of variables, constants, operators, and unit descriptors, with additional sub-attributes including scalar/vector/matrix for variables and local/global/discipline-specific labels for both constants and operators. Our extensive experiments show that state-of-the-art LLMs achieve an average of 20-60% accuracy under in-context learning and 50-60% accuracy with fine-tuning, revealing a significant gap in their mathematical reasoning capabilities. STEM-POM fuels future research of developing advanced Math-AI models that can robustly handle math symbols.

## 1 Introduction

Large language models (LLMs) have demonstrated exceptional reasoning abilities across numerous fields [17, 13, 12, 25, 15]. With the increasing shift towards applying LLMs to complex tasks [6, 23, 39], the need for supplementary data beyond the general pre-trained datasets has become increasingly important. Among these, mathematical reasoning tasks [10, 18] have recently drawn the attention of several researchers [19, 2, 45, 28] (see Backgrounds in Appendix B). In particular, Part-of-Math Tagging [43], the mathematical analog to part-of-speech tagging [36] where mathematical tokens are classified according to a given taxonomy of attributes, continues to gain interest but lacks the foundational datasets that can support advanced NLP tasks [43, 38, 37]. In addition, integrating mathematical language into NLP models remains a substantial challenge [3, 29], especially in the realm of document parsing [8, 24, 44]. Traditional semantic parsing methods like LateXML [31] or arXMLiv [22] often fall short when applied to math-rich documents, where precision and structured syntax are paramount [14, 32, 41]. These methods struggle to accurately perform pattern matching between abstract mathematical symbols and their corresponding XML tag notations. Similarly, recent advanced LLMs, such as ChatGPT [26], also face difficulties in understanding and reasoning with abstract mathematical symbols due to their contextual polymorphism [35] (as Figure 3 shown).

38th Conference on Neural Information Processing Systems (NeurIPS 2024).

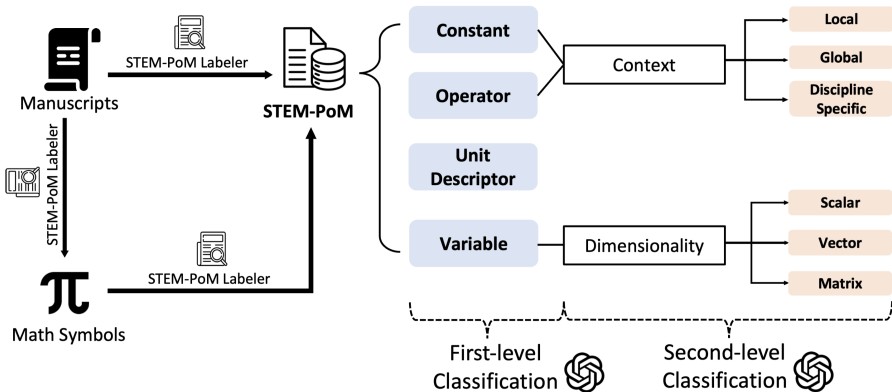

Figure 1: The overall pipeline for constructing the STEM-POM dataset. We extract math symbols with corresponding text information to formulate the dataset. Each math symbol is initially classified into one of four primary categories based on its definition. Then, the symbol is further categorized into secondary categories by the context in which it appears or by the symbol's dimensionality. An LLM is evaluated via the first-level and second-level classification tasks.

For example, in the linear equation: $y = mx + p$, $y$ is categorized as a variable. Whereas in the cross-entropy loss function: $\mathcal{L}(x, y) = -\sum_{i=1}^{N} x_i \log(y_i)$, the symbol $y$ represents the fixed target labels, which is considered a constant for a given dataset. Without the corresponding contextual information of a mathematical symbol, LLMs are unable to distinguish between different attributes of the symbol and cannot effectively process related mathematical reasoning tasks. Thus, tagging math symbols within domain-specific contexts is essential for language models.

In this paper, we introduce a novel benchmark dataset, **STEM-POM**, designed to evaluate the reasoning capabilities of language models on mathematical symbols across different domains. The STEM-POM dataset consists of 2,109 instances extracted from a random sampling of over 10,000 arXiv manuscripts, which are math-rich documents spanning domains such as Mathematics, Physics, Chemistry, and more. We provide a mathematical symbol for each dataset instance, its order in the document, its main and sub-level attributes, and the corresponding text or expression from the original arXiv paper containing the symbol. Each mathematical symbol in the dataset is classified according to two levels of attributes [42]. The first-level attribute categorizes the symbol as variable, constant, operator, or unit descriptor. The second-level attribute further classifies the symbol into one of six types based on its first-level category: scalar, vector, matrix, local, global, or discipline-specific. Figure 1 illustrates the dataset's category distribution. To further enrich the STEM-POM dataset with additional arXiv manuscripts and other math-rich document resources, we also design the **STEM-PoM Labeler**, a feasible method for assisting dataset generation by automatically searching, extracting, and recording hand-labeled mathematical symbols and their corresponding context from long-text documents.

We conduct thorough experiments on the STEM-POM dataset to assess the mathematical reasoning abilities of seven open- and closed-source language models, including LSTM [11], Mixtral-8x7B [20], Llama2-13B [40], Llama3-80B [9], Claude-3.5-sonnet [4], GPT-3.5, and GPT-4o [1] with various prompting and fine-tuning strategies. The experimental results indicate that STEM-POM distinguishes the performance levels across different LLMs, offering insights into the mathematical symbol reasoning abilities of these models. In addition, we investigate and analyze the influence of context length on the ability of language models to understand mathematical symbols.

## 2   STEM-POM Dataset

### 2.1   Data Annotation

**Source of Mathematical Symbols.** The first crucial step in constructing the dataset is selecting high-quality mathematical symbols. For STEM-POM, we primarily collect these symbols from two sources: 1. *Public math-symbol datasets*, where we directly utilize candidate mathematical symbols

| Statistic | Number |
|---|---|
| **Total Symbols** | **2,109** |
| **Unit Descriptor** | **129** |
| **Constant** | **384** |
| - Local | 171 |
| - Global | 121 |
| - Discipline Specific | 92 |
| **Operator** | **363** |
| - Local | 181 |
| - Global | 105 |
| - Discipline Specific | 77 |
| **Variable** | **1,233** |
| - Scalar | 601 |
| - Vector | 599 |
| - Matrix | 33 |
| Avg symbols per article | 4.7 |
| Avg tokens per sentence | 31.8 |
| Avg tokens per math symbol | 1.07 |

Table 1: STEM-POM Dataset Statistics

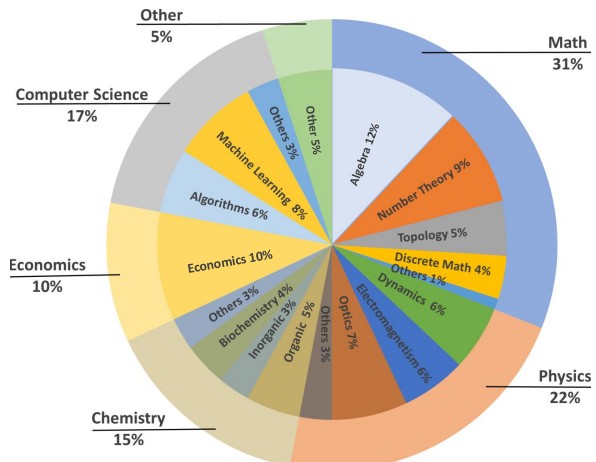

Figure 2: Discipline Distribution from Source ArXiv

from the mathematical token definition extraction benchmark, MTDE [14]. 2. *Raw ArXiv papers* [7], where we apply the STEM-PoM Labeler to identify and extract mathematical symbols from the ArXiv dataset. We include a detailed description of each source dataset in Appendix A.2.

**Dataset Construction.** After obtaining the mathematical symbols, we categorize each symbol into different attributes and assign corresponding information to construct the STEM-POM dataset. Specifically, we first extract the file name and symbol order for each mathematical symbol. Then, for each symbol, we extract the contexts in which the symbol appears, using several predefined lengths. Following this, we manually classify each symbol into four primary categories: Variable, Constant, Operator, and Unit Descriptor. For Variable, Constant, and Operator, we further categorize them into sub-level categories. The variable is classified as Vector, Scalar, or Matrix, while Constant and Operator are categorized as Local, Global, or Discipline-Specific. Table 2 outlines the overall dataset structure. We manually examine each entry of the dataset thoroughly to ensure its robustness and correctness. We provide a detailed explanation of the dataset structure in Appendix A.3 and the definitions of each level's attributes in Appendix A.4. Additionally, the STEM-PoM Labeler is described in Appendix A.5.

## 2.2 Dataset Statistics

We summarize the key statistics of our dataset in this section. Table 1 presents the categorical statistics, including the math symbols along with their first- and second-level attributes. The distribution of Variables, Constants, Operators, and Unit Descriptors is 58.5%, 18.2%, 17.2%, and 6.1%, respectively. In addition, Figure 2 illustrates the discipline distribution of the source arXiv papers. Our dataset covers mathematical symbols from various fields, including Mathematics, Physics, Chemistry, Economics, Computer Science, etc.

| File Name | Symbol Order | Symbol | Main Attribute | Sub Attribute | Related Contents |
|---|---|---|---|---|---|
| 9509/adap-org9509001.html | 0 | $f$ | Constant | Global | ...$1/f$ noise was discovered... |
| 9509/adap-org9509001.html | 1 | $\Delta$ | Operator | Global | ...can be quantified by studying the displacement $\Delta X$ |
| 9509/adap-org9509001.html | 2 | $X$ | Unit Descriptor | - | ...can be quantified by studying the displacement $\Delta X$ |
| 9509/adap-org9509001.html | 3 | $t$ | Variable | Scalar | ..after t steps, we can... |
| ... | ... | ... | ... | ... | ... |

Table 2: STEM-POM Dataset Structure

| Models | Context Length | Overall | Variable | Constant | Operator | Unit Descriptor |
|---|---|---|---|---|---|---|
| LSTM | One Sentence | 18.7% | 24.5% | 13.2% | 10.3% | 27.1% |
| | Ten Sentences | 22.6% | 28.1% | 16.8% | 15.5% | 30.2% |
| | Full Manuscript | - | - | - | - | - |
| Llama2-13B | One Sentence | 36.8% | 24.1% | 39.3% | 41.4% | 42.7% |
| | Ten Sentences | 42.7% | 35.6% | 39.8% | 46.9% | 48.5% |
| | Full Manuscript | 45.9% | 38.2% | 42.8% | 50.1% | 52.4% |
| Mistral-8x7B | One Sentence | 47.3% | 38.5% | 41.7% | 52.9% | 56.2% |
| | Ten Sentences | 49.8% | 41.8% | 45.9% | 58.6% | 56.7% |
| | Full Manuscript | 53.6% | 45.7% | 48.9% | 61.4% | 58.2% |
| Llama3-80B | One Sentence | 48.9% | 41.3% | 44.6% | 48.5% | 61.5% |
| | Ten Sentences | 53.0% | 44.8% | 48.8% | 54.7% | 63.7% |
| | Full Manuscript | 51.7% | 42.7% | 43.2% | 55.2% | 65.8% |
| Claude3.5-Sonnet | One Sentence | 63.7% | 58.6% | 62.5% | 65.7% | 67.8% |
| | Ten Sentences | 65.9% | 61.3% | 64.3% | 67.9% | 70.2% |
| | Full Manuscript | 66.7% | 62.9% | 65.8% | 68.6% | 69.3% |
| GPT-3.5 | One Sentence | 56.8% | 51.5% | 53.5% | 59.4% | 62.4% |
| | Ten Sentences | 58.7% | 54.5% | 53.6% | 61.3% | 65.1% |
| | Full Manuscript | 60.6% | 57.2% | 56.6% | 63.2% | 65.2% |
| GPT-4o | One Sentence | 64.9% | 60.5% | 64.2% | 64.9% | 70.1% |
| | Ten Sentences | 67.4% | 63.7% | 66.1% | 66.4% | 73.5% |
| | Full Manuscript | 68.5% | 64.2% | 67.8% | 68.1% | 73.8% |

Table 3: First-level classification accuracy with various context lengths.

| Models | Variable | | | Constant | | | Operator | | |
|---|---|---|---|---|---|---|---|---|---|
| (Vanilla) | Scalar | Vector | Matrix | Local | DS | Global | Local | DS | Global |
| LSTM | 13.8% | 15.1% | 17.2% | 19.2% | 17.8% | 22.2% | 16.6% | 11.3% | 14.6% |
| Llama2-13B | 27.3% | 24.4% | 21.8% | 33.6% | 31.5% | 33.6% | 32.4% | 28.3% | 32.7% |
| Mistral-8x7B | 36.9% | 35.8% | 21.6% | 34.8% | 31.2% | 37.8% | 36.4% | 34.8% | 35.7% |
| Llama3-80B | 38.2% | 34.1% | 26.7% | 37.6% | 35.2% | 36.1% | 39.1% | 32.3% | 40.2% |
| Claude3.5-Sonnet | 53.2% | 49.7% | 55.8% | 55.9% | 53.1% | 49.6% | 56.3% | 52.2% | 55.9% |
| GPT-3.5 | 44.5% | 45.8% | 48.3% | 48.5% | 42.9% | 44.3% | 48.4% | 43.5% | 49.7% |
| GPT-4o | 54.6% | 51.3% | 58.6% | 58.4% | 54.1% | 56.2% | 60.5% | 57.3% | 58.5% |

Table 4: Second-level classification accuracy with full manuscript input (Ten-sentence input for LSTM). We abbreviate "Discipline Specific" as "DS".

# 3 Experiments

## 3.1 Setup

**Models.** To thoroughly evaluate our dataset across models with varying parameter sizes, we utilize the following models: LSTM framework [11], Llama-2-13B [40], Mixtral-8x7B-v0.1 [20], and GPT-3.5-turbo-0125 [1].

**Evaluation Metrics.** We apply the *Precision Accuracy* as our metric for the mathematical symbol classification task, the metric can be formulated as: $Precision\ Accuracy = \frac{Number\ of\ correct\ predictions}{Total\ number\ of\ samples}$

**Training & Inference Details.** We evaluate several models under both pre-training and fine-tuning settings. Specifically, we train an LSTM model with varying layers and apply the LoRA method [16, 47], a PEFT technique, to GPT-3.5. We evaluate other models in the in-context learning setting. Appendix C provides the training and model parameter details.

## 3.2 First-Level Classification Results.

Table 3 presents the accuracy results for different models across varying context lengths. The result shows that the small-parameter-size language model such as the LSTM struggles with lower accuracy, achieving between 18.7% and 22.6%. In contrast, larger models, such as Llama2-13B and Mistral-8x7B, show marked improvements as context length increases, with Mistral-8x7B reaching up to 53.6% on the full manuscript. In addition, Claude3.5-Sonnet achieves comparable performance with GPT-4o across all context lengths, with accuracy consistently above 63.7% and up to 66.7%. GPT-based models exhibit stronger performance overall, with GPT-3.5 achieving between 56.8% and 60.6%. GPT-4o further improves across all context lengths, outperforming other models with an overall accuracy of 68.5% with the full manuscript input. We observe that the performance gap between models remains consistent as context length increases. For instance, GPT-4o outperforms Llama3-80B by 16.0%, 14.4%, and 16.8% for context lengths of one sentence, ten sentences, and the full manuscript, respectively. This consistent performance gap suggests that larger models with more pre-trained knowledge, such as GPT-4o and Claude3.5-Sonnet, exhibit superior scalability with longer contexts. These models are able to more effectively leverage extended context lengths to distinguish between mathematical symbols and other nuanced elements in the input prompts. On the other hand, the overall performance gain from increasing context length is more pronounced in smaller models, such as Llama2-13B and Mistral-8x7B, which have less pre-trained knowledge. These models benefit more from extended context as they rely on additional information to compensate for their limited pre-training. Larger models like GPT-4o and Claude3.5-Sonnet, which come with extensive pre-trained knowledge, show relatively smaller performance gains as context length increases.

## 3.3 Second-Level Classification Results.

Table 4 shows second-level classification accuracy with full manuscript input. In this experiment, we assume that the model got the first-level classification correct. From the results, LSTM performs poorly, with an accuracy as low as 11.3% for predicting the DS. Larger models, like Llama2-13B and Mistral-8x7B, improve performance, especially in classifying Constants (up to 37.8%). Llama3-80B shows moderate improvements, with 40.2% accuracy for Global Operators, indicating reasonable capabilities in operator classification tasks. Claude3.5-Sonnet and GPT-3.5 show further improvements, particularly in Global Constants and Operators classification. GPT-3.5 achieves 48.5% accuracy for Local Constants and 49.7% for Global Operators. Lastly, GPT-4o provides the highest accuracy overall, reaching 60.5% for Local Operators and 58.6% for Matrix classification. By horizontally comparing the same model performance on different sub-attribute classifications, we find that the attribute Constants are generally easier to classify compared to Variables and Operators across all sizes of models, as seen by the overall higher accuracy in Constant-related tasks. However, Matrix and DS classification continue to present challenges, even for the largest models, indicating that certain structures or content types within manuscripts remain difficult to categorize accurately at the sub-attribute level.

**Overall,** performance across all models on both first-level and second-level classification tasks shows a clear trend of improvement with increasing context length, highlighting the importance of context for accurately classifying mathematical symbols. Additionally, both small and large-size language models show a relatively higher accuracy in identifying Unit Descriptors and Operators compared to Variables and Constants, indicating that symbols with more distinct contextual or syntactical patterns are easier for models to classify. Through the above results, we aim to gain insights into the extent to which different category attributes of mathematical symbols influence LLMs' understanding of math-rich documents by correctly classifying the symbols in real-world scenarios. We leave additional experiments in Appendix D.

## 4 Conclusion

In this paper, we introduce STEM-PoM, a comprehensive benchmark for evaluating language models' mathematical reasoning abilities to classify math symbols from scientific texts. The dataset includes over 2,000 math instances sourced from ArXiv papers. Extensive experiments show that the best-performing model, achieves only 73.8% and 60.5% for first and second-level Part-of-Math Tagging classification accuracy, highlighting the challenge of extracting and categorizing mathematical symbols from large text corpora.

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

# A  STEM-POM Dataset Supplementary Materials

## A.1  Frequency Analysis on math symbols

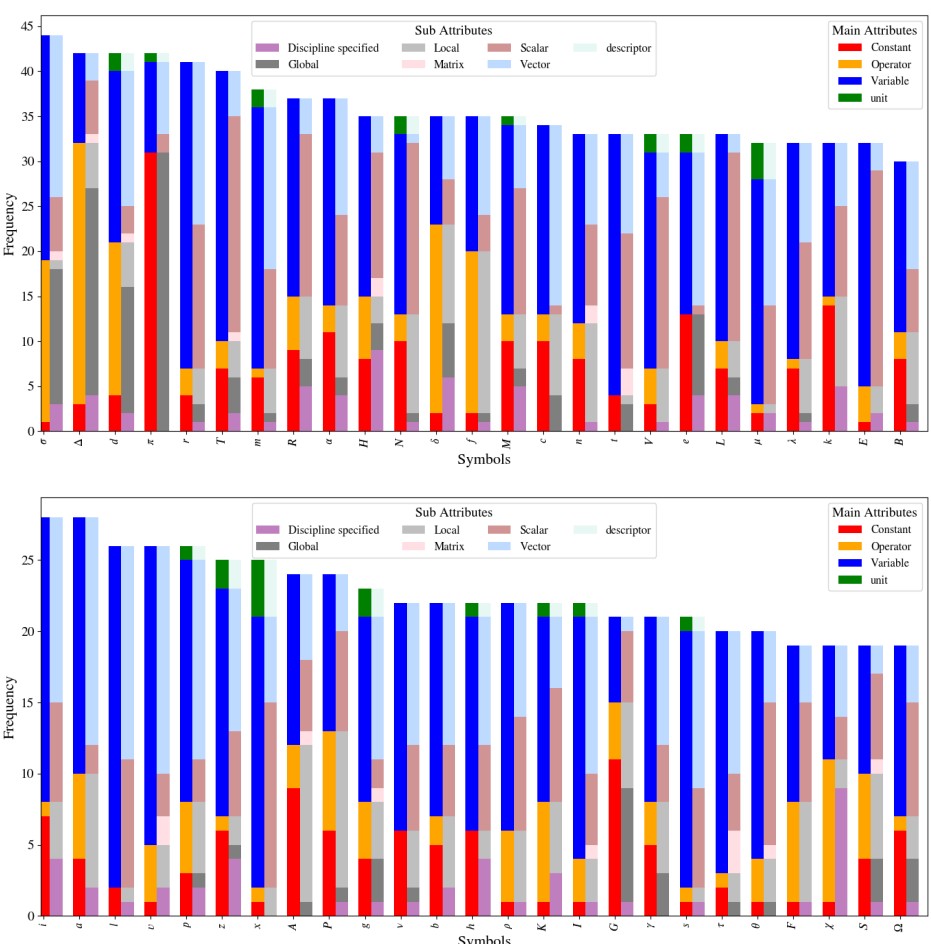

Figure 3: Total frequency (show-up times) of the top-50 mathematical symbols in the STEM-POM. This illustrates the contextual polymorphism of a single mathematical symbol, i.e. it belongs to multiple different attribute categories depending on the related context or mathematical expression.

## A.2  Source Dataset

**MTDE** [14] contains approximately 10,000 entries of mathematical symbol names along with their defined contexts. Each entry includes a 'short' definition and a 'long' definition. A short definition is a single-word definition, while a long definition consists of one or more words. The data was collected through random sampling from mathematical and scientific arXiv preprint manuscripts, covering a broad range of disciplines such as Physics, Computer Science, and Biology. For pre-processing, we ensured that the candidate data was generated via a corpus crawler and subsequently pruned and cleaned manually.

**ArXiv Paper Dataset** [22] contains 1.7 million arXiv articles, spanning a wide range of disciplines, including Mathematics, Physics, Chemistry, Economics, and Computer Science. We randomly sample 10,000 articles from this raw dataset and manually ensure that each manuscript is math-rich, containing numerous mathematical expressions and symbols. For pre-processing, we utilize the STEM-PoM Labeler to extract these symbols along with their surrounding context, ensuring that the data is representative of real-world mathematical usage across various scientific fields. Additionally,

the extracted symbols and contexts are systematically cleaned and structured to facilitate further classification and analysis.

## A.3 Dataset Definitions in Table 2

**File Name:** This attribute serves as a reference point, indicating the source of the file. Specifically, it denotes the arXiv article from which the dataset extracts its contents.

**Symbol Order:** This component records the sequence in which mathematical symbols appear within the article. By capturing the ordinal position of each symbol, we facilitate a structured analysis of the symbols' progression and their contextual relationships within the document.

**Symbols:** This field encapsulates the mathematical symbols themselves, predominantly consisting of Greek letters, albeit inclusive of additional characters. The precise documentation of these symbols is paramount for the subsequent analytical processes.

**Main and Sub Attributes:** These attributes categorize each mathematical symbol into specific classes, delineating a hierarchical structure within the dataset. This classification scheme is vital for understanding the symbols' roles and relationships within the mathematical discourse.

**Related Contents:** This segment comprises the words or sentences surrounding each symbol, embodying a critical resource for our model training. The contextual information surrounding each symbol is indispensable, as it imbues our models with a deeper understanding of each symbol's application and significance within the mathematical narrative.

## A.4 First-Level and Second-Level Attributes Definition

**Constant:** A value that does not change in a mathematical expression.

**Local Constant:** Constant that is specific to a particular system or model, such as the gravitational constant in a simulation of a specific planetary system.

**Global Constant:** Constant that is applicable in all contexts, like the speed of light in a vacuum.

**Discipline-specified Constant:** Constant that applies to particular fields of study, for instance, Planck's constant in quantum mechanics.

**Operator:** A symbol that operates on one or more operands.

**Local Operators:** Operator that is applied in a localized or specific context within a discipline, like a self-defined operation in matrix processing.

**Global Operators:** Operators that is used broadly across different disciplines, like the addition or multiplication operator.

**Discipline-specified Operators:** Operator that is unique to certain fields, such as the Hamiltonian operator in quantum physics.

**Variable:** A symbol that represents an unknown or changeable quantity in a mathematical expression.

**Scalar:** A quantity that has only magnitude, no direction.

**Vector:** A quantity that has both magnitude and direction.

**Matrix:** A rectangular array of numbers or symbols arranged in rows and columns.

## A.5 STEM-PoM Labeler

During the dataset construction, a pivotal step involves the meticulous annotation of each mathematical symbol with corresponding tags. This process, inherently labor-intensive and repetitive, necessitates a systematic approach to mitigate the workload and facilitate collaboration among the research team members. To address these challenges, we developed a labeling pipeline designed to streamline the dataset construction process. The UI design is shown in figure 4. The functionalities are delineated below:

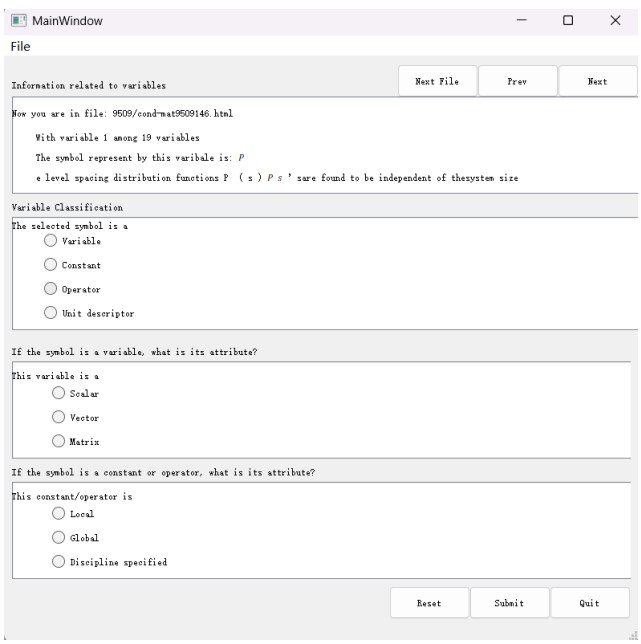

Figure 4: The UI Design of STEM-PoM Labeler

**File Reading:** We initiate the data importing operation progress by importing files from the designated arXiv folder, ensuring a structured and accessible repository of mathematical documents for subsequent processing.

**Symbol Identification and Contextualization:** For each file, we enumerate and display essential information: the current file being processed, the total number of symbols within, the sequence number of the current symbol, the graphical representation of the symbol, and the contextual content surrounding the symbol. This feature aids in providing a comprehensive overview and facilitates accurate symbol annotation.

**Annotation Interface:** We then present a user-friendly interface offering a set of predefined tagging options for each symbol. Through the designed interface, we easily select the most appropriate tag from these options, standardizing the labeling process and enhancing the consistency of the dataset.

**Data Recording:** Upon the selection of a tag for a symbol, We record this association by appending a new line to the dataset, capturing the symbol along with its assigned tag. This systematic data recording ensures the integrity and scalability of the MTCE dataset.

**Dataset Evaluation:** After constructing the dataset, we manually evaluate the quality and applicability of the annotated data. Specifically, we process the evaluation process through the following steps: Consistency Check, Inter-annotator Agreement, Statistical Analysis, and Benchmark Testing.

## B Backgrounds

**Part-of-Math (PoM) Tagging :** The part-of-math tagging task draws inspiration from similar tagging tasks such as part-of-speech tagging [36]. In the PoM context, the goal is to label individual mathematical tokens or expressions in math formulas with their corresponding mathematical roles. Such a task is essential for enabling a deeper semantic understanding of mathematical content by machines. Several datasets or benchmarks have been developed for the part-of-tagging task, but there also remain several challenges. Abdou [43] collects mathematical content, such as formula representation and tagging for specific mathematical formula translations and verifications, including converting formulae into semantic LaTeX or testing with tools like CAS (Computer Algebra Systems). However, this focus on structured and narrow formula translations does not align with the broader, more diverse text-based tasks required to assess NLP models, due to the lack of scalability features in the collected math symbols. Ruocheng [37, 38] recently evaluated the potential of leveraging

LLMs for automated annotation and Part-of-Math tagging of math symbols. However, their PoM tagging was conducted on the Digital Library of Mathematical Functions (DLMF) [27]. Since the source of math symbols is only one manuscript, the mathematical tokens collected only have a single classification type and are self-consistent. In contrast, our dataset incorporates the inherent messiness of published literature across several STEM subjects, where these domain-specific math symbols can have multiple classifications or meanings depending on the discipline and related context information.

**Large Language Models:** Pre-trained large language models (LLMs) have become a cornerstone in modern NLP [34, 46]. These models, which assign probabilities to word sequences by decomposing the probability of a sequence into the product of conditional probabilities of subsequent tokens, have evolved significantly over time. Early approaches were based on N-gram models, but with the advent of distributed word embeddings [5, 30], neural language models gained prominence. The scalability and performance improvements introduced by these models, along with the availability of vast textual data, have enabled the unsupervised pre-training of LLMs. These models, often referred to as foundation models [33, 23], can then be fine-tuned on smaller, task-specific datasets to adapt them for various downstream applications. For STEM-POM, we apply one traditional sequence-based NLP model, LSTM [11], and several recent LLMs for our dataset evaluation.

## C   Additional Experiment Setups

**Training Details** In our experiments, we train an LSTM with varying numbers of layers for the mathematical symbol classification tasks. For LLMs, we choose GPT-3.5 and apply a common parameter-efficient fine-tuning (PEFT) method, LoRA [16], to evaluate the model precision performance. We split our dataset into 80%/10%/10% for training/validation/testing sets.

**Model Parameters** For the LSTM model, we use different layer sizes from {128, 256, 512, 1024}. The hidden state size is set to 256, the learning rate is set from {0.1, 0.01, 0.001}, the training epoch is 5, and the batch size is 16. We utilize the Adam optimizer [21]. For GPT-3.5 fine-tuning, we use the GPT-3.5-turbo-0125 model version and set the training epoch to 3. For LoRA fine-tuning, we set the LoRA rank to 32, batch size to 32, weight decay to 0.01, dropout to 0.1, and learning rate to $1e^{-4}$.

## D   Additional Experiments

### D.1   Fine-tuning on STEM-POM

| Context Length | Overall | Variable | Constant | Operator | Unit Descriptor |
|---|---|---|---|---|---|
| *Vanilla Inference* | | | | | |
| One Sentence | 56.8% | 51.5% | 53.5% | 59.4% | 62.4% |
| Ten Sentences | 58.7% | 54.5% | 53.6% | 61.3% | 65.1% |
| Full Manuscript | 60.6% | 57.2% | 56.6% | 63.2% | 65.2% |
| *LoRA Fine-tuned* | | | | | |
| One Sentence | 67.4% | 64.8% | 67.5% | 71.4% | 66.1% |
| Ten Sentences | 66.9% | 65.4% | 66.6% | 71.3% | 64.5% |
| Full Manuscript | 62.2% | 58.4% | 62.2% | 65.1% | 63.2% |

Table 5: First-level classification with various context lengths on GPT-3.5 and fine-tuned GPT-3.5.

Table 5 shows the comparison result on main attributes between fine-tuned and directly vanilla-referenced GPT3.5. Notably, the fine-tuned GPT-3.5 model achieves an accuracy of 67.4% in the one-sentence context. However, its performance diminishes as the context length increases, with a noticeable drop to 66.9% for ten sentences and further down to 62.2% for full manuscript-length context. The decreasing trend suggests that while the fine-tuning process improves performance for shorter contexts, the model's ability to handle longer contexts is hindered.

The vanilla inference results also show a similar pattern of improvement with context length, but the gap between fine-tuned and vanilla inference narrows as the context length grows. For instance, the difference in overall accuracy between fine-tuned and vanilla models is 10.6% for one-sentence contexts but only 1.6% for full manuscripts.

Overall, the diminishing return for fine-tuned models with longer contexts indicates that fine-tuning amplifies sensitivity to the introduction of noisy or less relevant information when longer contexts are involved. The observation also could point to challenges in the fine-tuning process for long-context LLMs, which require more refined techniques to handle context length effectively.

## D.2   Ablation Study

| Model size(layers) | Variable | Constant | Operator | Unit Descriptor |
|---|---|---|---|---|
| 128 | 24.5% | 13.2% | 10.3% | 27.1% |
| 256 | 28.7% | 17.9% | 15.7% | 32.5% |
| 512 | 34.2% | 23.2% | 24.9% | 40.0% |
| 1024 | 46.5% | 35.9% | 44.2% | 51.3% |

Table 6: LSTM first-level classification accuracy based on different model sizes

| Context Length | Variable | Constant | Operator | Unit Descriptor |
|---|---|---|---|---|
| One Sentence | 24.5% | 13.2% | 10.3% | 27.1% |
| Five Sentence | 26.3% | 15.6% | 14.1% | 29.2% |
| Ten Sentence | 28.1% | 16.8% | 15.5% | 30.2% |

Table 7: LSTM first-level classification accuracy based on different input context lengths.

**Model Performance vs Model Size**   Table 6 presents the classification accuracy of an LSTM model for first-level classification across different model sizes, ranging from 128 to 1024 layers. Note that we set the input context length to be one sentence. The results show a clear positive correlation between the model size and classification accuracy across all four categories. For the smallest model (128 layers), the accuracy ranges from 10.3% for the Operator class to 27.1% for the Unit Descriptor class. As the model size increases, the performance improves notably, with the largest model (1024 layers) achieving a relatively high-performance gain in accuracy, ranging from 35.9% for the Constant class to 51.3% for the Unit Descriptor class. The most substantial improvements are observed in the Operator category, where accuracy increases from 10.3% for 128 layers to 44.2% for 1024 layers. These results suggest that larger model sizes are more effective in capturing complex patterns.

**Model Performance vs Data Input Lengths**   Table 7 displays the classification accuracy of an LSTM model across varying input context lengths across four categories. A trend of increasing accuracy can be observed as the input length increases. For instance, in the Variable category, the accuracy increases from 24.5% for one sentence to 28.1% for ten sentences. Similarly, for the Constant category, accuracy rises from 13.2% for one sentence to 16.8% for ten sentences. The Operator category shows a modest increase from 10.3% to 15.5% as the input length expands. Finally, for the Unit Descriptor category, accuracy grows from 27.1% to 30.2%. These results suggest that longer input data contributes to improved classification accuracy.

