# OpenReview forum: "STEM-PoM: Evaluating Language Models Math-Symbol Reasoning in Document Parsing"
_NeurIPS.cc/2024/Workshop/MATH-AI — MATH-AI 24_

### Official Review · Reviewer_TyKf · 2024-10-03
**A Review of STEM-PoM: Evaluating the Mathematical Reasoning of Language Models with STEM-POM**

**Rating:** 8
**Confidence:** 4

**Review:**

The paper "STEM-POM: Evaluating Language Models Math-Symbol Reasoning in Document Parsing" presents a novel benchmark dataset, STEM-POM, designed to evaluate the reasoning capabilities of language models (LLMs) on mathematical symbols within scientific texts. The authors aim to address the gap in LLMs' ability to understand and interpret abstract mathematical symbols in math-rich documents, a crucial aspect for advancing AI in STEM fields.

#### Quality:
The paper is well-structured and provides a comprehensive overview of the challenges faced by LLMs in mathematical reasoning. The introduction clearly outlines the motivation behind the study, and the methodology for dataset creation is detailed and robust. The experiments are thorough, covering a range of models and settings, and the results are presented with clarity.

#### Clarity:
The paper is generally clear, with a logical flow from the introduction to the conclusion. The authors provide sufficient background information and context for readers unfamiliar with the specific challenges of mathematical reasoning in NLP. However, some sections, particularly those involving technical details of the dataset construction and experimental setup, could benefit from additional clarification or simplification for broader accessibility.

#### Originality:
The introduction of the STEM-POM dataset is a significant contribution to the field, as it addresses a specific and underexplored area of LLM capabilities. The focus on mathematical symbol reasoning within the context of scientific documents is novel and timely, given the increasing application of LLMs in STEM fields.

#### Significance:
The findings of this paper are significant for the development of more advanced LLMs capable of handling complex mathematical reasoning tasks. The benchmark dataset provides a valuable resource for future research and development in this area. The results highlight the current limitations of state-of-the-art models, emphasizing the need for further advancements in this domain.

#### Pros:
- **Novel Dataset**: The creation of the STEM-POM dataset fills a critical gap in the evaluation of LLMs' mathematical reasoning abilities.
- **Comprehensive Evaluation**: The paper conducts extensive experiments across multiple models and settings, providing a clear picture of current capabilities and limitations.
- **Relevance**: The focus on mathematical reasoning in scientific texts is highly relevant to the ongoing integration of AI in STEM fields.

#### Cons:
- **Complexity**: Some sections of the paper, particularly those detailing the dataset construction and experimental setup, are complex and may be challenging for readers without a strong background in the field.
- **Limited Model Performance**: While the paper highlights the limitations of current models, it does not propose specific solutions or directions for improving LLMs' mathematical reasoning capabilities.

---

### Official Review · Reviewer_bjAw · 2024-10-07
**Well-described benchmark with potential limitations in model evaluation**

**Rating:** 6
**Confidence:** 4

**Review:**

Summary:
The paper introduces a new benchmark designed to evaluate language models' mathematical reasoning abilities to classify math symbols from scientific texts.

Strengths:
- Very good description of the benchmark (dataset statistics, discipline distribution, frequency analysis, etc.) and how it was constructed (sources, STEM-PoM Labeler, etc.).

Opportunities for improvements:
- It would be valuable to evaluate more recent and stronger models on STEM-PoM (GPT-4o, o1-preview, Llama 3, Claude, etc.) to better understand the challenge level for SOTA models.

Final remark:
The paper introduces a new benchmark with a clear description. However, I have some concerns regarding its impact, as newer models might already perform very well, limiting its usefulness for pushing the boundaries of mathematical reasoning in AI. For this reason, I opt for a rating of 6.

---

### Official Review · Reviewer_DYLo · 2024-10-07
**a novel benchmark dataset**

**Rating:** 6
**Confidence:** 4

**Review:**

This paper presents a novel benchmark dataset named STEM-POM, which is designed to assess the reasoning capabilities of language models when interpreting mathematical symbols within intensive scientific texts. The dataset consists of 2,109 instances covering fields such as Mathematics, Physics, and Chemistry. Each instance of the dataset includes a mathematical symbol, its position within the document, its primary and secondary attributes, and the surrounding context from the original paper.

1. By concentrating on mathematical symbols and their contextual interpretations across multiple domains, the dataset addresses a crucial gap in existing resources for testing language models' comprehension of abstract mathematical concepts.

2. The evaluations in the paper highlight the challenges that these models encounter in accurately interpreting mathematical symbols.

My query is whether your datasets can distinguish between current state-of-the-art models. I hypothesize that Claude 3.5 and GPT-4o are quite proficient at this type of task. Have you ever evaluated them? (I have not even seen the results of GPT4). I am apprehensive that your dataset may become obsolete as time passes.

---

### Decision · Program_Chairs · 2024-10-09

Accept